# Surveillance of U.S. Corporate Filings Provides a Proactive Approach to Inform Tobacco Regulatory Research Strategy

**DOI:** 10.3390/ijerph18063067

**Published:** 2021-03-16

**Authors:** Samantha Emma Sarles, Edward C. Hensel, Risa J. Robinson

**Affiliations:** 1Engineering Ph.D. Program, Rochester Institute of Technology, Rochester, NY 14623, USA; ses9066@rit.edu; 2Department of Mechanical Engineering, Rochester Institute of Technology, Rochester, NY 14623, USA; rjreme@rit.edu

**Keywords:** public health, decision-making, evidence, knowledge, translation, tobacco regulatory science, tobacco regulation, e-cigarette, e-cig, vape, Juul

## Abstract

The popularity of electronic cigarettes in the United States and around the world has led to a startling rise in youth nicotine use. The Juul^®^ e-cigarette was introduced in the U.S. market in 2015 and had captured approximately 13% of the U.S. market by 2017. Unlike many other contemporary electronic cigarette companies, the founders behind the Juul^®^ e-cigarette approached their product launch like a traditional high-tech start-up company, not like a tobacco company. This article presents a case study of Juul’s corporate and product development history in the context of US regulatory actions. The objective of this article is to demonstrate the value of government-curated archives as leading indicators which can (a) provide insight into emergent technologies and (b) inform emergent regulatory science research questions. A variety of sources were used to gather data about the Juul^®^ e-cigarette and the corporations that surround it. Sources included government agencies, published academic literature, non-profit organizations, corporate and retail websites, and the popular press. Data were disambiguated, authenticated, and categorized prior to being placed on a timeline of events. A timeline of four significant milestones, nineteen corporate filings and events, twelve US regulatory actions, sixty-four patent applications, eighty-seven trademark applications, twenty-three design patents and thirty-two utility patents related to Juul Labs and its associates is presented, spanning the years 2004 through 2020. This work demonstrates the probative value of findings from patent, trademark, and SEC filing literature in establishing a premise for emergent regulatory science research questions which may not yet be supported by traditional archival research literature. The methods presented here can be used to identify key aspects of emerging technologies before products actually enter the market; this shifting policy formulation and problem identification from a paradigm of being reactive in favor of becoming proactive. Such a proactive approach may permit anticipatory regulatory science research and ultimately shorten the elapsed time between market technology innovation and regulatory response.

## 1. Introduction

The popularity of electronic cigarettes (e-cigarettes) in the United States has led to a startling rise in youth nicotine use [1,2]. E-cigarette use has acted as a gateway to the use of other tobacco or tobacco-related products in non-established cigarette smokers [2]. Since their introduction to the market in 2004, electronic cigarettes have become popular among both cigarette users and nonsmokers. This may be partly because of a misconception that these products are a healthy alternative to cigarettes. While some studies report that e-cigarettes may lead to reduced harm compared to combustible cigarettes [3,4,5,6,7,8], others contrast with these findings [9,10,11,12]. Whether e-cigarettes will yield an overall positive or negative health impact compared to cigarettes remains unknown.

The Juul^®^ e-cigarette (Figure 1) rapidly came to dominate the U.S. market within a year after its 2015 release and led the way in the immense popularity of rectilinear form factor e-cigarettes and nicotine liquid filled pods. This e-cigarette represented a significant change from the prior “cig-a-like” and “pen style” devices which had thus far dominated the market, and ushered in the new era of compact high-nicotine concentration, discrete, easily concealed “pod style” devices. The Juul^®^ e-cigarette is composed of two primary subassemblies: the disposable reservoir and power control unit. The non-refillable reservoir contains nominally 0.7 mL of e-liquid composed of a mixture of propylene glycol, glycerin, nicotine and, in some cases, other additives. The top of the reservoir integrates a mouthpiece, which the consumer draws puffs through. The interior of the mouthpiece includes a sophisticated fresh air mixing chamber and flow conditioning pathways. The heating coil is located in the bottom of the reservoir and surrounds a wick to draw e-liquid into contact with the heating coil. The heated oven region of the reservoir is where aerosol is generated by mixing fresh air inlet from the bottom of the reservoir with vaporized e-liquid, and drawn upward through the outlet pipe into the mixing chamber. Two electrical terminals on the bottom surface of the reservoir (not shown) establish the electrical connection to the power control unit. The power control unit consists of an outside case, interior carriage, a microprocessor board, wire harness, 200 mAh lithium battery, and sensor gasket. Sides A and B of the disassembled PCU are illustrated in Figure 1. A four conductor USB wire harness connects the distal user-accessible USB port to the microprocessor board. The presence of four conductors suggests that the USB harness has potential both for power and data transfer (+V, ground, data + and data -). The microprocessor board includes battery charging circuitry, puff detection capability, coil voltage control, user feedback via light emitting diode, and data processing capability. The pressure sensor, used for puff detection, is evident on the opposite side of the microprocessor board. When assembled, the pressure sensor is encased by the sensor gasket to communicate the flow of fresh air. Two terminals connect the positive voltage and ground from the microprocessor board to the coil. Juul Labs™ Inc. has branded the reservoir as a Juulpod™, while many users refer to the disposable reservoirs as “pods.” It is common practice for users, manufacturers, and the literature to refer to the power control unit simply as the “battery”, obfuscating the potentially sophisticated user feedback and dynamic power control which could dramatically alter aerosol generation and emissions. 

It has been suggested that e-cigarettes may be a harm reduction product for established adult smokers [3,4,5,6,7,8], but long-term effects of the products and their consumables are not fully established. E-cigarettes are not harmless; they contain chemicals that may increase the risk of cancer and other diseases [13,14,15]. As of December 2020, the US Food and Drug Administration (FDA) has not approved any e-cigarettes as a cessation therapy. Despite this, several e-cigarette companies, including Juul Labs™, have advertised their product as cigarette replacements. The founders of Juul Labs™ designed their e-cigarette products with the intent of replacing the combustible cigarette [16]. 

The Juul^®^ e-cigarette has been criticized for its contribution to the youth nicotine epidemic in the United States [17,18,19,20]. While the relationship between social media and under-aged use of Juul^®^ products have been studied [21,22,23,24], there has not been independent, methodological investigation into the history of technology development and the corporate evolution of Juul Labs™. While a common view may be that the Juul^®^ e-cigarette rose to a position of market prominence in a fast and surprising manner, the information presented herein documents the decade-long journey from a graduate research project to becoming an “overnight sensation”. In retrospect, many leading indicators were available to see this looming change. This hindsight points to a significant gap in traditional tobacco regulatory research paradigms–emergent regulatory challenges can be anticipated through study of non-traditional research literature. 

This article presents a case study of Juul’s corporate and product development history in the context of US regulatory actions. The objective of this article is to demonstrate the value of government-curated archives as leading indicators which can (a) provide insight into emergent technologies and (b) inform emergent regulatory science research questions. 

## 2. Methods 

### 2.1. Sources of Data 

A variety of sources were used to gather data about the Juul^®^ e-cigarette and the corporations that surround it. Sources included government agencies, published academic literature, non-profit organizations, corporate and retail websites, and the popular press. 

Federal government agencies including the United Stated (US) Securities and Exchange Commission (SEC), the US Patent and Trademark Office (USPTO), US Center for Disease Control and Prevention (CDC), and the US Food and Drug Administration (FDA) were used to gather publicly available information and were considered authoritative sources. The Electronic Data Gathering, Analysis, and Retrieval (EDGAR) system search tool [25] was used to identify SEC filings related to the corporation. EDGAR is a structured search engine; once a corporate identifier was found that identifier was used to ensure SEC filings relevant to that corporation could be located. Trademark applications, patent applications, and issued patents were searched for in separate databases using the USPTO search tools. Keywords, such as corporation names, product names, and inventor names were used in the patent search tool. Owner and Registrant company names and product keywords including “Juul Labs”, “Pax Labs” and “Ploom” were used to search the USPTO’s Trademark Electronic Search System (TESS). Published patent applications were searched using the Patent Application Full-Text and Image Database (AppFT) and issued patents were searched using the Patent Full-Text and Image Database (PatFT) of the USPTO. The AppFT and PatFT databases were searched using the keywords “IN/Monsees AND IN/Bowen” to reflect that both principals were named among the inventors on the issued patents and published applications, respectively. The “Electronic Cigarettes” section of the “Smoking and Tobacco Use” module [26] of the CDC’s website was monitored between 2018 and 2020 for announcements and data relating to Juul Labs™ and its products. FDA announcements, guidance, and rulings relating to Juul Labs™ products were monitored by subscribing to FDA email notifications for any FDA activity relating to tobacco products, including announcements by the US FDA Center for Tobacco Products. CDC and FDA communications were included if it identified Juul Labs™ by name or was related to a Juul Labs™ product. 

The Delaware Division of Corporations [27] reports public information about companies incorporated in Delaware. Keywords “Juul,” “PAX,” and “Ploom” were used with DDC search tools to locate documents potentially related to Juul Labs™ and its affiliates. 

Published research literature related to Juul^®^ products was identified using Rochester Institute of Technology library search tools, subscription-based databases, and publicly available databases. The word “Juul” was used in conjunction with a filter that returned only peer reviewed journal articles. Articles were assessed for their relevance to this work by review of the abstract and conclusions. Articles deemed relevant were reviewed in more detail.

The Truth Initiative [28] and Tobacco Free Kids [29] are non-profit organizations that produce public information campaigns with facts about tobacco companies and nicotine use. Data from both organizations’ websites were captured and used in this work. While the Truth Initiative and Tobacco Free Kids are anti-smoking marketing tools, the campaigns cite government agencies and research institutions as well as their own surveys administered to youth. Data from both organizations were accepted as reliable and factual. 

The Juul Labs™ retail websites were monitored between 2018 and 2020. Changes in available products were documented. The internet archive tool *Way Back Machine* [30] was used to review the limited number of past versions of retail websites which had been archived. Two-source authentication was used for data captured from retail websites, both sources are cited in results.

In 2005, Adam Bowen and James Monsees presented their Master’s thesis for Stanford University’s product design program [16]. The presentation was uploaded to the Juul Labs™ YouTube channel on 27 February 2019. On 22 April 2019, the video was archived using youtube-dl [31], a program used to download videos from YouTube. The program was installed and run on a Linux machine. Once it was installed, the command ‘youtube-dl URL’ was run where “URL” was the YouTube video link. The video download was archived in a repository. It was attempted to obtain a copy of the written thesis. The authors names were used as keywords to search the ProQuest Dissertations and Theses database [32] and Stanford’s catalog and online institutional repository [33]. Emails were sent to the library at Stanford, the manager for mechanical engineering design at Stanford, and the Stanford Design Program Administrator. None of the contacted parties had provided a copy of the thesis at the time of manuscript submission.

Newspaper articles from CNBC, The Verge, TechCrunch, Forbes, The NY Post, and Bloomberg were used as sources for this work. Articles were located using keywords regarding specific events surrounding Juul Labs™. Data from reputable periodicals were used. Two-source authentication was used for most popular press articles. Both sources are cited in results where applicable. The search spanned the time from 1 January 2004 through 31 December 2020.

### 2.2. Data Sequencing and Event Counting 

Relevant facts and perspectives gathered from the sources were entered into a table of findings, placing events in chronological order with a concise description of each along with a short event title, classification (e.g., corporate event, milestone, trademark, patent application, issued patent, US Regulation, US State Regulation). Each document source was included as a single record in the findings table, while multiple data facts were extracted from each record. For example, each SEC filing was recorded as a single record in the findings table, but contained multiple facts including company name changes, executive officers and board membership, equity and debt placements, and private investor counts. Published patent applications were placed in the table based on their publication date, issued patents based on their issue date, and trademark applications based on their filing date. The title, abstract, claims, document images, and full text were assessed to determine if applications and patents were original or divisions or continuations of previous filings. Details of patent prosecution, issuance, division, continuation, extension, maintenance, and re-issue are not reported here. Data from published research articles were placed on the timeline based on the publication date of the paper. Changes to retail websites were given a general timeframe and each event on the timeline was approximate. News articles that described a change in the retail website, including change in product availability, were used to infer an approximate timeframe a change was made by the company. If a timeframe was given by the article, that date was used to place an event on the timeline otherwise the publication date of that article was used. Adam Bowen and James Monsees’ thesis work was placed on the timeline based on the time interval they were reported to have been in school for their master’s work. Data gathered from the presentation video were placed on the timeline in the year the presentation was given [17]. 

### 2.3. Data Interpretation 

After the data were compiled by date, each item was sorted into one of eleven categories: corporate events, influence, research literature, market, patents (issued), patent applications (published), milestones, trademark applications, product development, U.S. Regulation, and State Regulation. Regulations outside the USA were not included. While partial data were collected for U.S. State regulatory actions, state-level data were not presented herein, as it was found to be challenging to ensure their completeness, due to their dynamically changing nature. This paper presents and interprets data from the Corporate, Patent Application, U.S. Regulation, Milestones and Trademark Application categories. Data from the remaining categories are deferred to a future work. 

### 2.4. Data Exclusion

Trademark applications identified using TESS were disambiguated and trademarks or wordmarks not related to the field of use of electronic cigarettes and vaporizers were excluded. For example, the keyword “JUUL” was, at one time, associated with custom clothing made by a designer having the surname “Juul” and a company named “Arx Pax Labs” was disambiguated from “Pax Labs.” These unrelated search results were excluded from data reported here. The company names associated with trademark filings were searched using the SEC EDGAR search tool and a generic internet search engine. If no evidence could be found linking key persons or company addresses to Juul Labs™, those data were not considered for this work. Issued patents and published patent applications resulting from USPTO searches were reviewed to determine whether each patent or application was related to a Juul Labs™ product or if the reference was to Juul Labs™ by a third party as part of a prior art citation. Any documents which did not appear to be assigned to Juul Labs™ or any of its affiliates, are not presented in this work. Data from non-reputable sources, such as a blog, were not included in this work if the piece of data could not be authenticated by a second source. 

## 3. Results

Figure 2 depicts the development of Juul Labs’™ technology and corporation, and its predecessor companies in the context of United States e-cigarette regulation between 2004 and 2020. The timeline is presented in the context of three major “eras” of focus: “MS Thesis. Tobacco”, “Product Development focused on Loose Leaf Tobacco and Marijuana”, and “Product Development Focused on Nicotine E-liquids.”

The nineteen “Corporate Filings and Events” illustrated in Figure 2 are composed of six DDC filings, ten SEC filings related to exempt offerings of securities, and three events documented through literature citations. While many activities occur within privately held companies which are never published, the DDC and SEC filings provide fixed points in time at which historical events related to the company can be documented. Appendix A: Corporate Filings and Events provides a detailed summary of all the key facts harvested from each of the nineteen documents. 

The twelve “US Regulation” results presented in Figure 2 reflect FDA and CDC notices, activity reported by the US Congress legislation tracker, the Federal Register, and published literature. While some “first of a kind” state regulatory actions are presented in the following text, state actions are not illustrated in Figure 2. Appendix A: US Regulation provides a summary key facts and dates associated with each regulation.

The sixty-four “Monsees and Bowen Patent Application” results presented in Figure 2 document the history of publication dates of each application. Patent applications are published by the USPTO one year after the application filing date, whether or not a patent has been issued. Patent prosecution often takes longer than a year, so applications are often the earliest published document which provide details regarding inventions and emergent technology. All 64 patent applications shown here included both Bowen and Monsees among the list of inventors. Appendix A: US Patent Applications provides a detailed summary of the patent applications. Of those 64 applications, there were 27 unique application titles, and the remaining 37 publications typically reflected divisions, continuations and subsequent prosecution of earlier applications. The application date of the 27 initial applications occurred one year prior to the entry on the time line. In many cases, that initial application was preceded by a provisional patent application, statutorily no more than one year preceding the full application. Of course, the inventive work had to occur prior to the application filings. Thus, it is reasonable to interpret each published patent application reflective of research and development occurring between 1 and 3 years earlier. 

Eighty-seven trademark applications were determined to be associated with Juul Labs, Pax Labs, and Ploom. Appendix A: US Trademark Applications provides a detailed summary of the trademark and wordmark applications, descriptions of goods and services, and maintenance status. Many of the applications consisted of word mark applications, with re-filings to address changes in the scope of goods and services associated with various marks. Only three of the seventy-four word mark filings are presented in Figure 2, in the interest of brevity, while all symbolic trademarks are shown in year of the initial filing. Cancellation of symbolic trademarks is indicated in the year of cancellation by virtue of a red circle overlaid on the trademark. 

At least fifty-five patents have been issued which are in some way affiliated with Ploom, Inc., Pax Labs, Inc., Juul Labs, Inc., Adam Bowen, or James Monsees. Of these, there were 23 design patents protecting ornamental design features of the companies’ products, with the remaining 32 utility patents protecting underlying technologies. Appendix A: US Patent Grants summarizes the titles, key dates, inventors and assignees of the protected inventions. 

Selected details of the corporate events, patent applications, US Regulations, and trademark applications depicted in Figure 2 are presented as a historical summary to facilitate subsequent discussion. Significant additional data are available in the Appendix A for readers interested in specific aspects of the companies’ research and development history. 

### 3.1. MS Thesis Era 

The first era, between 2004 and 2005, reflects graduate master’s degree collaboration by Juul Labs™ founders Adam Bowen and James Monsees. This collaboration resulted in the 2005 milestone event where they presented their idea for an electronic nicotine delivery system in their 2005 thesis presentation [16] at Stanford University, claiming that they set out to design for social change. They recognized that cigarette smoking had become socially delinquent and wanted to turn their tobacco product into a “luxury good”, not just a drug delivery device [16]. They conducted market research by asking cigarette smokers what they loved about smoking and characterized these as attributes their device must have. They also asked smokers what they did not like about smoking and characterized these items as qualities their device must not possess. The creators also avoided using nicotine iconography, a different approach than many e-cigarette makers [16]. 

### 3.2. Loose-Leaf Tobacco Era 

The second era, beginning in 2006 and continuing until 2013, saw the founders incorporate their first companies and focus much of their product development effort broadly in the field of loose-leaf tobacco and marijuana consumables.

Juul Labs™, Inc. and Ploom Investment, LLC filed incorporation papers on 12 March 2007 (DDC File Number 4315504) and 30 May 2007 (DDC File Number 4361140), respectively, with the Delaware Division of Corporations. 

The first joint patent application for a “Method and System for Vaporization of a Substance” (USPTO Patent Application US 2007/0283972 A1) by James Monsees and Adam Bowen was published on 13 December 2007, based on a utility patent application which had been filed on 11 July 2006, which in turn, relied upon the provisional application No. 60/700,105 filed on 19 July 2005. Since the first application claimed priority based on a provisional application from 2005, it is apparent the invention was related to the pair’s graduate studies. Next, Paax, Inc. filed as a Delaware Corporation on 29 January 2008 (DDC File Number 449618). 

On 15 April 2009 (and subsequent 20 April 2009 notice of action), the FDA ordered that a shipment of NJOY e-cigarettes be denied entry to the US under the Federal Food, Drug, and Cosmetic Act. NJOY had imported and distributed such products in the US since 2007. On 22 June 2009 the US Family Smoking Prevention and Tobacco Control Act was signed into law [32]. This law did not specifically include electronic nicotine delivery systems (ENDS). In response, NJOY filed suit against the FDA questioning whether “Congress has authorized the Food and Drug Administration (“FDA”) to regulate e-cigarettes under the drug/device provisions of the Federal Food, Drug, and Cosmetic Act (“FDCA”), 21 U.S.C. § 351 et seq., or under the Family Smoking Prevention and Tobacco Control Act of 2009 (the “Tobacco Act”), Pub. L. 111–31, 123 Stat. 1776”. On 7 December 2010, the federal court of appeals’ ruling (Sottera, Inc. dba NJOY v. FDA, 627 F.3d 891 [D.C. Cir. 2010]) [33] set a precedent, giving the FDA authority to regulate ENDS devices and consumables under the Tobacco Act and not under the FDCA. The court ruled that derivatives of tobacco were to be regulated as a “tobacco product” unless “marketed for therapeutic purposes,” in which case they would be regulated as drugs and/or devices [33].

While NJOY was engaged in the two-year court battle with the FDA, Juul continued developing innovative technology related to loose-leaf tobacco. Three additional patent applications were published in 2009, including “Aerosol Devices and Methods for Inhaling a Substance and Uses Thereof” (US 2009/0151717 A1) claiming priority of provisional application No. 61/014,690 filed on 18 December 2007, and “Method and System for Vaporization of a Substance” which consisted of two application publications (US 2009/0260641 A1 and US 2009/0260642 A1) as divisions of application No. 11/465,168 filed on 11 July 2006, which in turn claimed priority of the same provisional application No. 60/700,105 filed on 19 July 2005 described above. The ‘641 and ‘642 applications both listed Ploom, Inc. as the assignee. 

On 6 May 2011 Ploom, Inc. filed their first notice of exempt offering of securities with the US Securities and Exchange Commission for USD 3,330,163 in equity and options, having sold USD 3,230,163 to 5 investors as of the filing date (SEC AN 0001520049-11-000001). James Monsees and Adam Bowen were listed as the sole executive officers, with a board comprised of Bowen, Monsees, Tom Dykstra (Tao, LLC) and Riaz Valeni (Ploom Investment LLC). Thus, the founders accepted their first round of capital investment within five years of completing their graduate degree. This USD 3.3 million dollar investment set the tone for the company to pursue a growth arc more like a high-tech startup and unlike a traditional tobacco company. 

### 3.3. Nicotine E-Liquid Era

The transition from a focus on loose-leaf tobacco (Era 2) towards nicotine e-liquids (Era 3) appears to have begun mid-way through the year 2013 and included several important trademark and patent applications and investments. After a decade of studying vaporization of loose-leaf botanicals, Bowen and Monsees shifted their focus to nicotine salts, an e-liquid ingredient that supposedly gives a strong nicotine rush to users, vaporized in an elegant, sleek vaporizer that evokes a “ritual”, similar to that of a traditional cigarette [34,35]. 

Two patent applications titled “Low Temperature Electronic Vaporization Device and Methods” were published on 21 February 2013 (US 2013/0042865 A1) and 28 November 2013 (US 2013/0312742 A1) both claiming priority of provisional application No. 61/524,308 filed on 16 August 2011. These two applications referred to a device to *“generate an aerosol for inhalation by a subject heating a viscous material that can have a tactile response in the mouth or respiratory tract*.” These two documents disclose many novel features which would subsequently become associated with the innovative product known as the Juul^®^ electronic cigarette, almost two years later. 

The transition towards electronic cigarettes accelerated in 2014 when Japan Tobacco International [36] entered into an exclusive long-term cooperation agreement whereby JTI would commercialize PAX Labs’™ ENDS device to sell outside of the United States [36]. JTI acquired the Ploom product line, intellectual property related to the Ploom device, including the Ploom trademark, and a minority stake in PAX Labs™ [36,37]. Financial details of the agreement were not released. This transaction likely provided a significant infusion of working capital to the founders and probably generated a stream of cash flow; Bowen and Monsees could now afford to shift their focus from heat-not-burn tobacco products towards electronic cigarettes. Again, this rapid shift from one technical innovation to another one, is reminiscent of a high-tech startup company, not a traditional tobacco company. 

The year 2015 was a busy one, highlighted by the milestone event of Juul^®^ electronic cigarettes entering the US market. Pax Labs filed two exempt offerings of securities in 2015 (changing the name from Ploom Inc. to Pax Labs Inc.) on 7 April and 4 June 2015 (SEC AN 0001520049-15-000004 and 0001520049-15-000005, respectively). Adam Bowen and James Monsees were listed as the sole executive officers, with an expanded board comprised of Bowen, Monsees, Nicholas Pritzker (Tao, LLC), Riaz Valeni (Ploom Investment LLC) and Harold Handelsman (c/o PAX Labs). The first filing disclosed a USD 25 million equity round, with USD 6.5 million sold to 4 investors at the time of the filing and the amended filing, just two months later, increased the equity sought to USD 46,685,837 already sold to 54 investors. 

The application titled “Vaporization Device Systems and Methods” (US 2015/0208729 A1) was published on 30 June 2015 and referred to three earlier applications dating as early as 23 December 2013. This application described the essential embodiments of the Juul^®^ electronic cigarette and disclosed a sophisticated feedback system for coil temperature control, pressure sensor activation, charging circuitry, user feedback, coil and wick geometry, preferred materials, and detailed assembly instructions. Images from the ‘8729 application were extracted and annotated and are presented in the composite Figure 3. The numerical annotations reference the static description of the invention as disclosed in the application. It is instructive to compare the annotated photographs of Figure 2 against the patent application graphics of Figure 3. Virtually every novel aspect of the Juul^®^ e-cig introduced in 2015 is documented in this published application. It was clear, as early as June 2015, that this product contained significant advances in technology in comparison to other e-cigs in the market at the time. At the time this product was introduced to the market, it was unlike any of the cig-a-like or pen-style electronic cigarettes available. This patent application foreshadowed practically every technical innovation that would be pursued by the company through to the end of the decade and beyond. 

One of the relatively few issued patents not jointly identifying Bowen and Monsees as co-inventors was invented by Bowen and Xing, titled “Nicotine salt formulations for aerosol devices and methods thereof,” filed in October 2014 and granted in December 2015 [38] claiming priority of two provisional applications filed in 2013. The nicotine salt formula is one aspect of what Juul Labs™ advertises to make their product innovative. Nicotine salts are found in leaf-based tobacco. According to Juul Labs™ [35], this constituent creates a unique smoking experience, similar to smoking a combustible cigarette. In retrospect, the nicotine salt innovation would become the centerpiece of discussion about the youth nicotine epidemic and the influence of nicotine salts on the ability of electronic cigarette users to tolerate high nicotine concentration e-liquids. 

On 10 May 2016, six years after the Sottera, Inc. dba NJOY v. FDA ruling, the FDA published the rule “Deeming Tobacco Products To Be Subject to the Federal Food, Drug, and Cosmetic Act, as Amended by the Family Smoking Prevention and Tobacco Control Act; Restrictions on the Sale and Distribution of Tobacco Products and Required Warning Statements for Tobacco Products,” which asserted the agency’s regulatory authority over e-cigarettes. This rule, which had been originally proposed by the FDA on 25 April 2014, went into effect on 8 August 2016 and included the requirement of premarket data or substantial equivalency requests for all newly regulated products (such as e-cigarettes not already commercially marketed in the US as of 15 February 2007) [39]. Following this rule, regulation of e-cigarettes in the US began to increase. Meanwhile, 14 additional patent applications were published in 2016 and 2017, the details of which foreshadowed the evolution of Juul^®^ products. 

A potentially transformative patent application entitled “Calibrated Dose Control” (US 2016/0157524 A1) was published on 9 June 2016 and claimed reference to two previous unpublished applications filed as early as 5 December 2014. This application describes *“Methods and vaporizer apparatuses that estimate, measure and/or predict the amount of vapor and/or material (including active ingredients) released by the vaporizer apparatus…”* This application appears to be the first evidence of an invention to quantify and report the dose of substances, such as nicotine, delivered from an electronic cigarette to the mouth of the product user. The remaining seven applications published in 2016 represented continued prosecution, divisions and continuations of previously described applications. Word mark applications filed in 2016 provide hints into emerging technologies being investigated by Juul, which would not become publicly evident until two or three years later, when corresponding patent applications were eventually published, 12 months after the initial patent application filing date.

The company raised USD 200 million additional investment during 2016 and 2017. Pax Labs (formerly known as Ploom, Inc.) filed notice of exempt offering of securities with the SEC in 2016 for an offering amount of USD 29,999,997, and amended the filing to increase the amount to USD 54,999,995 in 2017. The second filing in 2017 was by Juul Labs™, Inc. (formerly Ploom, Inc. and Pax, Inc.) for an offering amount of USD 150,000,000. This investment was critical to the rapid growth of the company in expanding their US market position. By the end of 2017, Juul Labs™ led all U.S. e-cigarette sales capturing about 13% of the market [40]. The Juul^®^ e-cigarette had become “one of the fastest-growing consumer products in history” according to *Fast Company* [17]. When the founders of PAX Labs launched the Juul^®^ e-cigarette in 2015, they approached it as launching a new technology, not a drug delivery system. They used a traditional start-up narrative: “identify the needs, scale the solution” [17], which means the company was founded on novel innovation to fulfill specific consumer needs with the goal of making a rapid transition from a small start-up company to securing a major market position. 

Four novel patent applications were published in 2017, along with applications which appeared to be continued prosecution of earlier invention filings. All 2017 published applications centered on enhancing and protecting the product technologies making Juul’s products distinctive in the marketplace at the time. Building upon their 2015 word mark MAKE THE SWITCH TODAY, the company filed another word mark application for JUUL. MADE FOR SWITCHING (USPTO SN 87369494) on 13 March 2017, both of which signaled Juul’s intent to promote their product as an alternative to combustible cigarettes. 

The growing popularity of the Juul^®^ e-cigarette raised concerns among lawmakers, prompting the US government to act in late 2018 when the FDA commissioner proclaimed that youth e-cigarette use was at “epidemic proportions” [41]. In September 2018, the FDA began to issue thousands of warning letters and civil money penalty complaints to retailers who were illegally selling Juul^®^ and other e-cigarette products to minors [42,43]. The companies were given 60 days to demonstrate steps they would take to keep their products out of the hands of minors [19]. The FDA also requested information related to marketing practices from Juul Labs™ and several other e-cigarette manufacturers [19,42]. A month later, in October 2018, the FDA raided Juul Labs’™ headquarters in San Francisco with a surprise onsite inspection [19,44]. During this inspection, the FDA obtained thousands of documents from Juul Labs™ related to marketing [44]. The FDA was concerned about Juul^®^ products’ role in the rise of youth tobacco use [42]. Shortly after the FDA’s unannounced visit to Juul Labs’™ headquarters, the FDA gave the company and four other popular e-cigarette device makers an ultimatum. In November 2018, the FDA took the first large step in the regulation of electronic cigarettes by announcing a plan to limit the sale of flavored cartridge-based e-liquids to smoke shops and online [41]. The agency also released their plan to implement age-verification guidelines for online sales [18]. 

During 2018, Juul Labs™ appeared to implement a plan to pre-empt regulation that would likely affect sales. Juul Labs™ rebranded from a company characterized by bright colors, contemporary graphics, and young people using Juul^®^ e-cigarettes [45] to adopt a medicinal ambiance [46]. Juulpod™ packaging was altered to include a nicotine warning on the front and the nicotine strength listed in much larger, prominent font. Further, Juul Labs™ renamed many of their flavors, replacing friendly adjectives like “cool” with a name that describes flavor objectively. Juul^®^ Labs also deleted all content from their social media accounts and released TV commercials directed at an older audience, with an emphasis that the product is for established cigarette users who want to stop smoking [46]. In late October 2018, Juul Labs™ pulled their child-friendly flavored products, including all e-liquids with flavor additives other than tobacco and menthol flavor, from the shelves of physical retail stores, excluding 18+ smoke shops [47], noted as a milestone event on Figure 2. It should be noted that Juul’s decision to pull flavors in October 2018 preceded the formal FDA action in November 2018 which placed restrictions on the sale of flavored cartridge-based e-liquids. In the fourth quarter of 2018, Juul Labs’™ net revenue reportedly dropped for the first time in several quarters, during the same timeframe the FDA limited the sale of flavored e-liquid products [48].

Juul filed another exempt offering notice for an offering amount of USD 1,250,000,000 (with USD 650,000,000 sold to ten investors as of 19 August 2018, with the first date of sale on 26 June 2018) as equity (SEC AN 0001520049-18-000001). Juul was valued at an estimated USD 38 billion in late 2018 when the tobacco giant Altria invested USD 12.8 billion for a 35% stake in the company, announced on 18 December 2018 [49]. The details of the transaction between Altria and Juul were not disclosed. It was not clear what portion of the USD 12.8 billion investment went to buying out previous private shareholders, and what portion represented issuance of shares under previous SEC filings. Clearly, the Altria investment was larger than the sum of all SEC debt and equity placements identified herein. 

Twenty-two Juul patent applications were published in 2018. Most of these applications were related to continued prosecution of previous submissions regarding core technology of the company. Two applications published in 2018 are worthy of mention. The first was titled “Leak Resistant Vaporizer Device” (US 2018/0077967 A1) claimed priority of a previous application filed on 22 September 2016 and describes *“leak-resistant vaporizer cartridges and/or apparatuses adapted for use with oil-based vaporizable materials including cannabis oils …”* which suggests the era of cannabinoids may be resurrected in Juul’s future. The second application, “On-Demand Portable Convection Vaporizer” (US 2018/0000160 A1) was related to two earlier applications dating back to 16 June 2016, and describes *“… heating air drawn through an oven chamber to a predetermined or selectable vaporizing temperature to vaporize a material (e.g., loose leaf plant material, etc.) that is held in the oven chamber …”*. This application indicates continued interest on the part of Juul to develop technology for vaporizers of loose-leaf botanicals, hearkening back to the founders’ graduate studies. The 2018 word mark applications for DYNAMIC MODES (USPTO SN 88886595), and JUUL TALK (USPTO SN 87956869), could be interpreted as foreshadowing future corporate intentions. 

The 2019 Juul milestones of “introducing a reduced nicotine concentration e-liquid” to the market and “eliminating flavors” other than tobacco and menthol are shown in Figure 2. Both of these milestones could be reasonably attributed to the FDA regulatory actions during 2018. Two significant federal actions occurred in 2019, including addition of a prominent nicotine warning on the front of e-cigarette products, and the FDA issuance of a warning to Juul Labs™, Inc. for marketing unauthorized modified risk tobacco products. On 17 November 2019, Massachusetts became the first state to ban the sale of flavored e-cigarette products other than tobacco and menthol flavor [50], while hundreds of localities across the United States have since formed their own legislation restricting sale of flavored e-cigarette products [50].

In early 2019, as recent FDA regulations on e-cigarettes were being enforced, Juul Labs™ worked to expand its team. The company hired a former Cardinal Health executive to lead a team tasked with creating partnerships between Juul Labs™ and “health plans, providers, self-insured employers, and the public sector” [51]. The “enterprise markets team” included 17 employees as of March 2019. 

Juul Labs filed another notice of exempt offering of securities in the amount of USD 325,000,000 (with USD 325,000,000 sold to four investors as of the 8 August 2019 filing date with the first sale reported as 2 August 2019) as equity and debt (SEC AN 0001520049-19-000003). Less than two weeks later, Juul amended the securities filing to increase amount to USD 800,176,874 (with USD 785,176,874 sold to 14 investors as of the 30 August 2019 filing date) as equity and debt (SEC AN 0001520049-19-000004). This USD 800 million dollar investment, following on the heels of Altria’s USD 12.8 billion infusion suggests clear corporate intent to invest in a combination of research and development, infrastructure and distribution channels. 

Six patent applications were published in 2019. Of these, two applications present insights into technologies which were not readily apparent in Juul’s commercial products at the time. One important application titled “Electronic Vaporizer Sessioning” (US 2019/0158938) claimed priority of a previous application filed on 22 November 2011 and provides insight into Juul’s technology development plans for *“Devices, systems and methods for electronic vaporizer sessioning …”* They described a vaporizer device (such as an e-cigarette) connected to a user device (such as a mobile phone or computer) via the user interface of an application. The publication described how the software app could compare an individual’s actual usage with their preferences during a time interval, and determine settings for the vaporizer device aligned with the user’s preferences. This application foreshadows capability of the Juul product line enabling users to manage their level of product use.

Another significant application, “Vaporization Device Control Systems and Methods” (US 2019/0069599) described the underlying technologies which would enable the sessioning methods described in the previous paragraph. The application described *“… methods for controlling the power applied to a resistive heater of a vaporization device by measuring the resistance of the resistive heater at discrete intervals. Changes in the resistance during heating may be used to control the power applied to heat the resistive heater during operation…”* and vaporization devices configured to employ those methods. This document describes increasingly sophisticated feedback control logic allowing the power control unit to limit the rate and amount of aerosol generated in the oven. 

Juul filed a significant RACS trademark application on 6 August 2019, as shown in Figure 2 (USPTO SN 88568223). This trademark application described the goods and services as *“Promoting awareness of new authentication and identification technologies to prevent retail sale of nicotine products to underage consumers …”* and development of corresponding voluntary standards. This trademark filing appeared timely in light of recent regulatory actions and the national spotlight on the youth e-cigarette epidemic. It also provides the research community with a clear indication of potential future technology and market innovations by the company. In conjunction, word mark applications filed in 2019 claim goods and services related to downloadable software, remotely setting and adjusting vaporizer power parameters, *“transmitting and reporting information relating to the location, movement, proximity, departure and arrival of individuals and objects”* and technological innovations which might be forthcoming in future months and years. 

On 28 February 2020, the US House of Representatives passed a bill titled Protecting American Lungs and Reversing the Youth Tobacco Epidemic Act of 2020. If made law, the bill would revise *“requirements related to the safety, sale, and advertisement of tobacco products, including electronic nicotine delivery systems,”* including *“prohibit[ing] the use of flavored products in an electronic nicotine delivery system”* [52]. On 2 March 2020, the bill was received by the US senate and read twice. The US Congress bill tracker reported the bill died at the conclusion of the 116th congress (2019–2021), having never been passed by the US Senate. Meanwhile, New York, New Jersey, and Rhode Island passed their own statewide laws to ban flavors in 2020 [53].

Juul filed one exempt SEC offering during 2020 for USD 721,555,937 (with USD 721,555,937 sold to 20 investors as of the 7 February 2020 filing date, and the first date of sale reported as 3 February 2020) as a combination of debt and equity (SEC AN 0001520049-20-000001). As of this filing, K.C. Crosthwaite (who had spent most of his earlier career at Altria) had replaced Kevin Burns as CEO and board member and Saurabh Sinha replaced Timothy Danaher as Chief Financial Officer. 

Five applications published in 2020 are of particular note and demonstrate Juul’s continued technological development in the areas of smart electronic cigarettes and heat-not-burn products. The first application published in 2020 appeared on 2 January 2020 and was titled “Connected Vaporizer Device Systems” (US 2020/0000143 A1) and described emergent technology being invented by the company. The abstract is a compelling summary of what may be expected from future Juul products. The abstract states:


*“A vaporizer system may include a vaporizer device communicatively coupled with a user device configured to control the functions and/or features of the vaporizer device. The vaporizer device may serve as a replacement for traditional combustible cigarettes. Accordingly, the user device may be configured to collect usage data from the vaporizer device and generate recommendations to enhance and/or expedite the transition from traditional combustible cigarettes to the vaporizer device. For example, the user device may provide puff coaching to enable a more satisfying initial experience. Alternatively and/or additionally, the user device may recommend pod types and/or puff patterns that are associated with a reduction in overall intake.”*


Of particular interest are the statements regarding technologies designed to enhance or expedite users switching from combustible cigarettes and/or reducing overall intake. Whether the company seeks approval for future products as a tobacco cessation therapy remains to be seen, however, this application establishes a clear path forward for such innovation. 

This smart e-cigarette theme continued throughout 2020, including an application published on 12 November 2020, titled “User Interface and User Experience for a Vaporizer Device” (US 2020/0352249 A1) which describes *“… a device in communication with a vaporizer, can include one or more features related to control of functions and/or features of the vaporizer, identification of a cartridge and/or a vaporizable material in the cartridge, data exchange (either one-way or two-way) between a cartridge and a vaporizer with which the cartridge is engaged…”*


Concurrently with the development of smart e-cigarettes, Juul Labs remained focused on advancing technologies hearkening back to their corporate origins. The application titled “Cartridge-based Heat Not Burn Vaporizer” (US 2020/0037669 A1), assigned to Juul Labs, provides some indication of Juul potentially moving into the arena of heat-not-burn non-liquid vaporizable materials. The background, summary and description of the invention application focused primarily on loose-leaf tobacco and parts of tobacco leaves, with passing reference to additional broad categories of prescribed medicants and other plant-based smokeable materials such as cannabis, including *“solid (e.g., loose-leaf) materials, solid/liquid (e.g., suspensions, liquid-coated) materials, wax extracts, and prefilled pods (cartridges, wrapped containers, etc.) of such materials.”*

Several word mark applications were filed between 10 February and 14 December 2020. The mark BEACON (USPTO SN 90381115) describing goods and services related to *“downloadable software for use with oral vaporizers and electronic cigarettes”* takes on additional significance in the context of the RACS trademark file in 2019 and the several patents application published in 2020. The word mark BINGO (USPTO SN 88790625) described goods and services related to near field communication technology programming and reading devices–such as might be needed to incorporate radio frequency identification into future products. 

## 4. Discussion

The objective of this article was to demonstrate the value of government-curated archives in providing insight into emergent technologies and informing emergent regulatory science research questions. Our research lab engages in emissions characterization and topography behavior monitoring for emerging products. We, along with the scientific community, have been in a constant struggle to adapt our research methodologies to keep pace with the rapid innovation in the marketplace. We have long recognized that we have to focus our research not only on those tobacco products in the market today, but also predict those product innovations which will appear in the future. Technology surveillance as outlined here can be a valuable tool in shortening the time lag between the market introduction of new tobacco products and appropriate evidence-based regulatory response.

### 4.1. Traditional Public Policy Is Reactive

Public policy is often described as an iterative cycle of five steps including (1) problem identification and agenda setting, (2) policy formulation, (3) decision making and formal policy adoption, (4) policy implementation, and (5) policy assessment or evaluation. Before a policy is created by a governing body, some person or event must bring an issue to the forefront and garner the attention of public authorities. Policies cannot be formulated or adopted by decision makers until after an issue has been identified. Then, and only then, can policies be put into practice. When policies are in operation, a formal assessment process (for example a sunset clause on a policy) may be triggered, or a new issue may arise which was not foreseen by the policy, to begin the cycle again. 

### 4.2. Government Archives Provide Hindsight

A classic example of reactive public policy is the sequence of events surrounding the deeming rule as illustrated by Figure 2. The Tobacco Act was passed on 22 June 2009, and was cited almost immediately by NJOY in their legal response to the April 2009 FDA action taken against them under the Food, Drug and Cosmetic Act (FDCA). The resultant 2010 court decision led, four years later, to FDA proposing the deeming rule in April 2014. The final deeming rule was not published until after public comment and response period in May 2016, and did not go into effect until August 2016. Thus, the steps of policy formulation and policy adoption spanned more than seven years, and all aspects of the pre-market tobacco application (PMTA) requirements were still not implemented as of December 2020–eleven years after the initial attempted FDA action under the FDCA. All regulatory agencies, such as the FDA, are limited in their actions by the statutory authority vested in them, inherently leading to significant delays between problem identification and a policy implementation. 

Conversely, companies often anticipate regulatory actions before they are officially enacted. A good example of this can be seen with Juul’s preemptive October 2018 flavor milestone, presumably in an attempt to stave off and potentially minimize the breadth of the FDA’s November 2018 formal action. 

### 4.3. Government Archives Provide Foresight

It is no surprise that government archives provide historical context to events in the past; we study history to learn its’ lessons in an effort to avoiding repeating past mistakes. However, it is also true that historical government archives can provide insight into emergent regulatory science research questions. 

When a closely held company, such as Juul Labs™, raises billions of dollars in debt and equity, this is an indicator they have the financial wherewithal to conduct meaningful product research and development (R&D). When investments are accompanied by trademark and patent applications, we can make reasonable predictions about corporate R&D which prompt regulatory science research. 

Under-age use of tobacco products has been, and continues to be, of critical interest to regulators. It is clear that Juul Labs™ is investing in, developing, and preparing to market technologies to curb under-age youth use of tobacco products. Consider the RACS, BEACON and BINGO trademark applications filed during 2019, which pre-dated the 2020 Youth Tobacco Epidemic Act, in the context of patent applications filed that same year. It is clear that companies, including Juul Labs™, will soon have the ability to assist with limiting tobacco sales to underage youth at the point of sale. This knowledge establishes the firm premise for urgent regulatory research questions, such as: “Do electronic point of sale interventions effectively reduce access to tobacco products by under-age youth?” If this research can be conducted quickly and answered affirmatively, a scientific foundation may be established for a regulation requiring point of sale access control technology for all tobacco products.

### 4.4. Regulatory Science Research May Become Proactive

Technologies to limit the dose of nicotine or other substances delivered to a user are coming. The invention disclosed in Juul Labs’™ patent application “Calibrated Dose Control” (US 2016/0157524 A1) demonstrated companies have the ability to incorporate dosage limits into electronic cigarette products. Research such as [54,55] lays the foundation for establishing a connection between the nicotine and Total Particulate Matter (TPM) dose delivered to the mouth of the user. Given that companies will be able to modulate nicotine dose, and we can correlate nicotine consumption with TPM exposure from a mechanistic perspective, then we can open a pathway to better understanding public health effects of proposed regulatory action related to nicotine dose control. This observation suggests immediate research is needed to establish whether, and under what conditions, there exists a nicotine dose threshold during a specified time interval which inhibits addiction. The traditional approach of regulating one product characteristic (such as e-liquid nicotine concentration) and not addressing a closely related product characteristic (maximum power dissipated in the coil of an e-cigarette) creates a variety of “pathways” or “loop holes” that companies can “design around.” A more effective approach might be to regulate the desired outcome. Now that an invention has been disclosed by at least one company to control the dose of nicotine delivered to the mouth of a user, we may choose to regulate the outcome measure of primary interest: nicotine dose. If nicotine addiction is deemed a public health issue then “dose delivered” may be a valid product characteristic to regulate. If, instead, TPM exposure is deemed the outcome measure of primary interest, then combining the company-demonstrated nicotine dose control with mechanistic research relating nicotine yield to TPM yield [54,55], then “TPM delivered” may be a valid product characteristic to regulate. This approach could foster a fundamental shift in tobacco regulation. Instead of regulating product characteristics (coil resistance, voltage, temperature, nicotine concentration) which engineers can readily design around, we can regulate outcome measures (already demonstrated by corporate disclosures) directly. Companies cannot argue “we don’t have the ability to control dose” when inventions have been filed which clearly document such capability. We should begin conducting research now to facilitate future regulations requiring dose control technology in combustible tobacco, heat-not-burn products, and vaping products. 

## 5. Conclusions

Research funding agencies and their review panelists, particularly those focused on regulatory science, should acknowledge the probative value of citations to patent, trademark, and SEC filing literature as a means of identifying emergent research questions. 

Funded research on emergent research questions may significantly reduce the time lag between problem identification, policy formulation, and policy adoption. 

The methods presented herein permit the first two steps in the policy cycle, problem identification and policy formulation, to be undertaken proactively—dramatically shortening the elapsed time between problem identification and policy implementation. 

## Figures and Tables

**Figure 1 ijerph-18-03067-f001:**
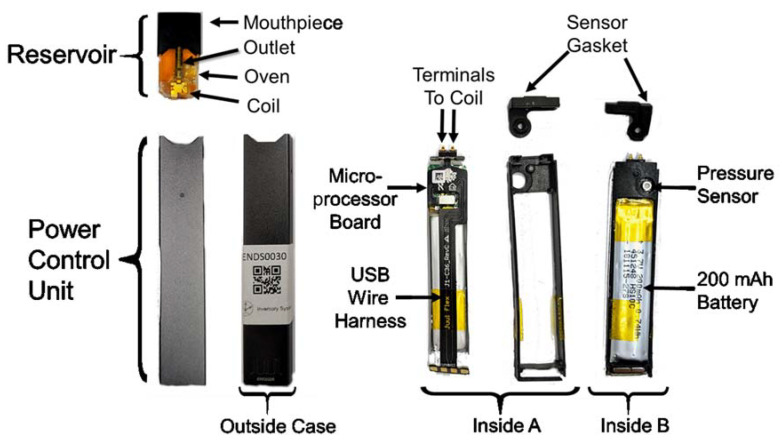
Annotated photographs of a Juul^®^ electronic cigarette and subcomponents, purchased from a retail establishment in 2019. Several features of this product represented significant technological innovations in comparison to contemporary electronic cigarettes at the time the product was introduced. Photos by Respiratory Technologies Lab, Rochester Institute of Technology.

**Figure 2 ijerph-18-03067-f002:**
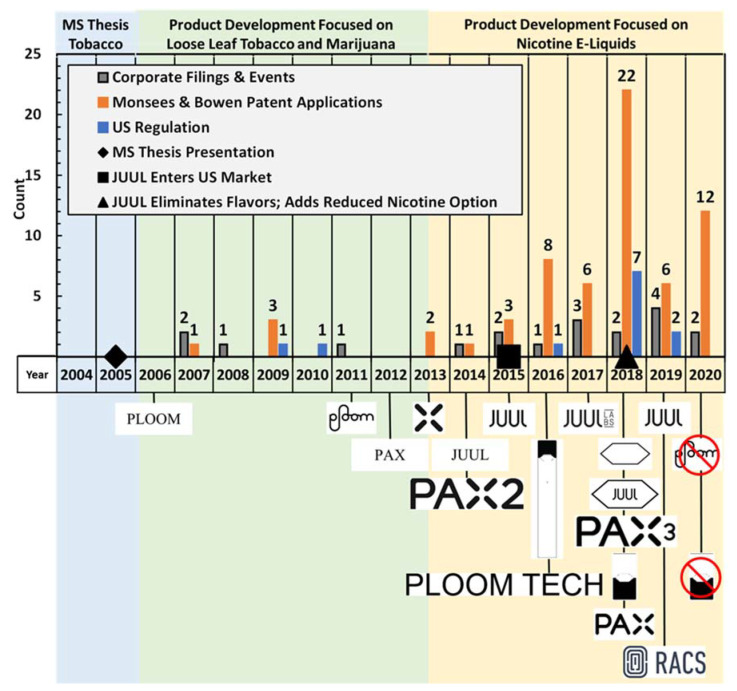
Chronological depiction of Juul Labs’™ technology and corporate development in the context of United States e-cigarette regulation between 2004 and 2020. All trademarks presented are owned by Ploom, Inc., Pax Labs, Inc., or Juul Labs, Inc.

**Figure 3 ijerph-18-03067-f003:**
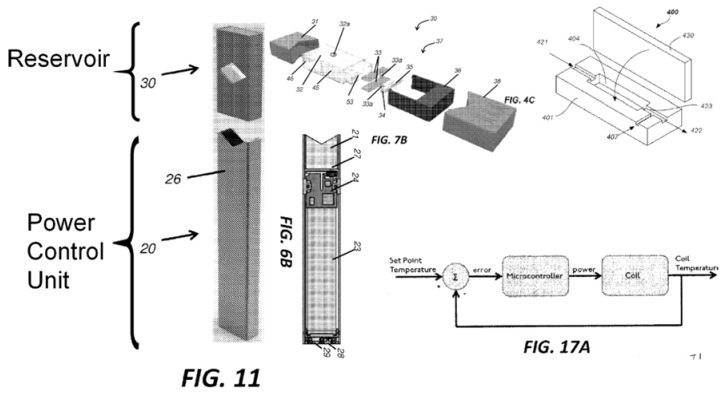
Annotated composite of multiple figures presented in US Patent Application 2015/0208729 A1 published on 15 June 2015. This document illustrated the dramatic innovations present in the novel Juul electronic cigarette, and visible in products dissected years later.

## Data Availability

The Appendix A provide the cross-references needed to access publicly archived datasets associated with Corporate Filings and Events, U.S. Regulation, U.S. Patent Applications, U.S. Trademark Applications, and U.S. Patent Grants, respectively.

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
