# Peer review of "Surveillance of U.S. Corporate Filings Provides a Proactive Approach to Inform Tobacco Regulatory Research Strategy"

_ijerph, 2021, doi:10.3390/ijerph18063067_

Round 1

Reviewer 1 Report

Introduction

  1. I suggest deleting the whole part about how that particular e-cig is built. Please address only main differences (if there are any) to other e-cigs available.
  2. Fig. 1 seems also not necessary.
  3. Lines 80-87. I do not fully agree. First of all, numerous papers revealed significantly lower toxicity of e-cigs compared to conventional ones. Of course, e-cigs are still harmful - at least due to the sole nicotine content. However, what should be clearly stated here is that the probable toxicity highly depends not on the e-cig itself, but on the liquid composition e.g. nicotine content and also some additional substances.

Results

  1. Fig 2. - I suggest deleting the lower half - it is neither informational nor easy to understand. Optionally, please prepare a clearer version with a better caption. In its current form, this figure is to hard to fully understand.
  2. Whole "results" section should be rewritten. Most of the data should be presented in the form of tables. The text should be clearer and should present a history. Currently, it lacks consistency and seems a bit chaotic. Moreover, there is really hard to see the point of some long parts of it.
  3. Consider plotting some parts of data e.g. company value or related statistics against some e-cig-related facts.
  4. Moreover, please focus on those facts that are anyhow connected to public health issues. Please refrain from mentioning all of the business details that have no real and significant value from the public health perspective. IJERPH focuses on matters related to public health, not the business. Business-oriented things are important only as long as they are relevant to public health.
  5. Finally, please organise results into subject-oriented sub-sections

Discussion

  1. Given the chaotic nature and complexity of results, it is hard to say whether Discussion is anyhow appropriate. I suggest, first rewriting results as previously suggested, then creating a suitable discussion that orbits around the main theme of results.
  2. Lines 773 - 775. It seems like a considerable part is missing here ("...").

General

  1. The article is too long and too chaotic. It requires significant corrections, in fact, is should be mostly rewritten. First of all, please identify the central topic and build the whole history around it.
  2. Currently, I feel like introduction and discussion belongs to one paper, but the "results" belong to yet another one!
  3. Finally, from a health perspective. It seems to me that authors mostly focus on effects related to cancer and pulmonary problems, completely omitting those related to e.g. cardiovascular risk or fertility issues.

Author Response

Authors’ Response to Review Report 1

Thank you for reviewing our manuscript and providing your useful notes and suggestions. Your feedback is valuable to us and has been considered while revising our manuscript. Responses to each of your points are provided, along with an indication of how the manuscript has been revised. We hope that our edits address each of the items raised and have improved the manuscript.

Reviewer Comments Related to: Introduction

Reviewer Comment: I suggest deleting the whole part about how that particular e-cig is built. Please address only main differences (if there are any) to other e-cigs available.

Authors’ Response: The Juul e-cigarette represented a significant design variation from all previous electronic cigarettes, which had heretofore been disposable cig-a-like and pen-style devices. The product which is the focus of this article ushered in an entire new category of devices and was dramatically more sophisticated in engineering design than any prior e-cig. This technically innovative product, in later years, was cited as the driving force behind the resurgence in youth nicotine addiction in the USA. We have added text to the introduction which highlights the importance of the technical innovations present in the Juul e-cig, which would go on to have dramatic public health impacts.

Reviewer Comment: Fig. 1 seems also not necessary.

Authors’ Response: We have added text describing the novelty of this device, and expanded the figure caption to state “Several features of this product represented significant technological innovations in comparison to contemporary electronic cigarettes at the time the product was introduced “ This product was the landmark product which introduced the entire category of “pod style” electronic cigarettes to the global market. We hope the revised text makes it clear that many tobacco product innovations throughout the decade were enabled by the introduction of this novel device.

Reviewer Comment: Lines 80-87. I do not fully agree. First of all, numerous papers revealed significantly lower toxicity of e-cigs compared to conventional ones. Of course, e-cigs are still harmful - at least due to the sole nicotine content. However, what should be clearly stated here is that the probable toxicity highly depends not on the e-cig itself, but on the liquid composition e.g. nicotine content and also some additional substances.

Authors’ Response: We respectfully disagree that the “… toxicity highly depends not on the e-cig itself, but on the liquid composition…” The design of the device has a direct impact on the operating temperature of the coil, which in turn plays a significant role in the generation of decomposition byproducts arising from degradation of the e-liquid and exacerbated by materials choice in the device. The e-cig devices have become highly sophisticated power management devices with potential to dramatically increase or decrease public health harm. Regulatory public health scientists have rightly spent years studying the toxicity of e-liquids. It is our contention, supported by the documents cited in this manuscript, that e-cigarette manufacturers are investing hundreds of millions of dollars in research and development to increase the efficacy of inhalant delivery. The corporate filing, patent applications and trademark applications documented herein demonstrate an intentional design evolution spanning two decades of technological innovation. Numerous articles have indeed demonstrated the potentially positive public health impacts of reduced harm from e-cigs, offset by a concern of increased youth nicotine use and rising concerns about product mis-use and EVALI lung injury.

Reviewer Comments Related to: Results

Reviewer Comment: Fig 2. - I suggest deleting the lower half - it is neither informational nor easy to understand. Optionally, please prepare a clearer version with a better caption. In its current form, this figure is to hard to fully understand.

Authors’ Response: Thank you. We realize the figure contains highly dense information and was difficult to digest. We have revised the lower half of Figure 2 in an effort to simplify it, and remove references to month within each year. Now, we show these events only at the level of year.

Reviewer Comment: Whole "results" section should be rewritten. Most of the data should be presented in the form of tables. The text should be clearer and should present a history. Currently, it lacks consistency and seems a bit chaotic. Moreover, there is really hard to see the point of some long parts of it.

Authors’ Response: Thank you for your input. We have removed all text from the results which is not directly supportive of the comments made in the Discussion and Conclusion sections. We have reduced the level of financial details related to the securities offerings, and have added phrases to provide context about the relevance or significance of those filings.

Reviewer Comment: Consider plotting some parts of data e.g. company value or related statistics against some e-cig-related facts.

Authors’ Response: We debated adding cumulative dollar value of investment to the figure, but were concerned this would detract from the figure clarity. The intent of Figure 2 is indeed to related three aspects of company development (Corporate investment, patent applications and trademark applications) in the context of government issued regulations.

Reviewer Comment: Moreover, please focus on those facts that are anyhow connected to public health issues. Please refrain from mentioning all of the business details that have no real and significant value from the public health perspective. IJERPH focuses on matters related to public health, not the business. Business-oriented things are important only as long as they are relevant to public health.

Authors’ Response:  The premise of our article is that business details are, in fact, critical predictors of emergent public health issues. The goal of this article is to demonstrate that corporate investment and R&D publications provide predictive value and can establish the premise for emergent public health research questions. We document the six year elapsed time between the first FDA action against the NJOY e-cigarette manufacturer until the adoption of the deeming rule. During that six year period, the electronic cigarette industry remained largely unregulated. And, it was precisely during that time interval which Juul conducted most of their product innovation. By the time most FDA regulations were imposed, they were already obsolete. We believe this is clearly relevant to public health.

Reviewer Comment: Finally, please organise results into subject-oriented sub-sections

Authors’ Response:  We have shortened the results section, while retaining the three main subsections used in the original manuscript. We have attempted to focus each main paragraph of the results on a single topic or a single year of the product development history. We hope the sequential-in-time presentation of results is sufficiently clear.

Reviewer Comments Related to: Discussion

Reviewer Comment: Given the chaotic nature and complexity of results, it is hard to say whether Discussion is anyhow appropriate. I suggest, first rewriting results as previously suggested, then creating a suitable discussion that orbits around the main theme of results.

Authors’ Response: We have reorganized the discussion into four key points using subsections. Without introducing a lot of additional text, we have tried to make a one to one connection between each of the four discussion points and the results presented in the timeline.

Reviewer Comment: Lines 773 - 775. It seems like a considerable part is missing here ("...").

Authors’ Response: Thank you. We have corrected this omission.

Reviewer Comments Related to: General

Reviewer Comment: The article is too long and too chaotic. It requires significant corrections, in fact, is should be mostly rewritten. First of all, please identify the central topic and build the whole history around it.

Authors’ Response: We hope that the re-write of the results and discussion section, and inclusion of four supplemental data tables, will address this concern. We have tried to focus on the central theme of connecting corporate documents to emerging public health research questions, with the goal of decreasing the time to implement effective tobacco product regulations.

Reviewer Comment: Currently, I feel like introduction and discussion belongs to one paper, but the "results" belong to yet another one!

Authors’ Response: We have tried to reiterate the primary theme in the Introduction, Methods, Results, Discussion and Conclusion. It is our intent that the details presented in the Results section provide evidence in support of the statements made in the Discussion and Conclusion.

Reviewer Comment: Finally, from a health perspective. It seems to me that authors mostly focus on effects related to cancer and pulmonary problems, completely omitting those related to e.g. cardiovascular risk or fertility issues.

Authors’ Response: Thank you. While the focus of this paper is not to document the breadth of all possible health issues associated with tobacco use, we certainly agree that the impacts are broad. We hesitated to expand this section of the Introduction, given the length of the article.

Reviewer 2 Report

In an era, where being proactive helps societies stay alive and refrain from coronavirus casualties, this paper gives a new perspective on how the companies legislation footsteps can help science proact at the future problems rather than react at their consequences.

I believe this work, although rather exhausting sometimes, because of the detailed analysis of the company’s adjustments to new policies and state restrictions, can provide an example of how this knowledge can be used by research facilities to design experiments that will provide solutions to problems that have not yet been shown to the market. This approach can finally help universities and other research facilities to connect with society and design specific experimental approaches which will provide solutions and give new perspectives at the industry section.

As a biologist, and not a lawyer, I found the last part (the Discussion and Conclusions) of the paper far more interesting than the Results. I would advise the authors to shorten the Results Section and describe the most important changes and decisions taken throughout this decade, summarizing the rest. They can add a section as a Supplementary Material for what they consider less important and keep in the main context only the milestone achievements of the strategy of the company.

I also have some minor changes that need to be checked out:

Line 27: “the years 2004 thought 2020” do the authors mean “the years 2004 through 2020”

Line 165: “None of the contacted parties provided a copy of the thesis” I believe that PhD Theses are uploaded in the online institutional repositories and can be found electronically or at the library of the University. In any case, if information was not available, it has no meaning referring to it in the main text.

Line 261: Two dots at the end of the sentence.

Line 369: “This application discloses a two compartment vaporizer wherein the material being vaporized in each compartment could be the same or different, and the vaporization temperature of each compartment could be the same or different” I think this sentence needs to be rephrased into “ ..where the material as well as the temperature can differ at each of the two compartments”.

Line 782: I think a “What” is missing between “insight into” and “technology” in order for the sentence to make sense.

Author Response

Authors’ Response to Review Report 2

Thank you for reviewing our manuscript and providing your useful notes and suggestions. Your feedback is valuable to us and has been considered while revising our manuscript. Responses to each of your points are provided, along with an indication of how the manuscript has been revised. We hope that our edits address each of the items raised and have improved the manuscript.

Reviewer Comment: In an era, where being proactive helps societies stay alive and refrain from coronavirus casualties, this paper gives a new perspective on how the companies legislation footsteps can help science proact at the future problems rather than react at their consequences.

Authors’ Response: Thank you. This was exactly our goal for the article. We appreciate your following suggestions and comments as a mean of helping us better achieve that goal.

Reviewer Comment: I believe this work, although rather exhausting sometimes, because of the detailed analysis of the company’s adjustments to new policies and state restrictions, can provide an example of how this knowledge can be used by research facilities to design experiments that will provide solutions to problems that have not yet been shown to the market. This approach can finally help universities and other research facilities to connect with society and design specific experimental approaches which will provide solutions and give new perspectives at the industry section.

Authors’ Response: Thank you for your thoughtful comments. We are sorry the manuscript was exhausting to read … as you point out, harvesting knowledge from these non-traditional data sources may permit public health oriented scientific research to become proactive. We hope our edits, particularly, to the Results section, may reduce the burden on the reader.

Reviewer Comment: As a biologist, and not a lawyer, I found the last part (the Discussion and Conclusions) of the paper far more interesting than the Results. I would advise the authors to shorten the Results Section and describe the most important changes and decisions taken throughout this decade, summarizing the rest. They can add a section as a Supplementary Material for what they consider less important and keep in the main context only the milestone achievements of the strategy of the company.

Authors’ Response: Thank you. We have added supplemental data tables for the corporate filing documents, patent applications and the trademark applications. We have removed a significant amount of text from the results and made reference to the supplemental data tables.

Reviewer Comment: I also have some minor changes that need to be checked out:

Authors’ Response: We appreciate the care you have taken in reading the manuscript, and have attempted to address each item raised.

Reviewer Comment: Line 27: “the years 2004 thought 2020” do the authors mean “the years 2004 through 2020”

Authors’ Response: Thank you for catching this error. We have corrected it.

Reviewer Comment: Line 165: “None of the contacted parties provided a copy of the thesis” I believe that PhD Theses are uploaded in the online institutional repositories and can be found electronically or at the library of the University. In any case, if information was not available, it has no meaning referring to it in the main text.

Authors’ Response: We thought it might be interested to the reader to know that, despite our attempts to secure this academic information which is usually readily available, we were unable to locate the thesis referred to by the founders. We have no way of knowing if a thesis was never actually produced, or whether it was embargoed and removed from circulation by the university.  Per your suggestion, we have removed this sentence.

Reviewer Comment: Line 261: Two dots at the end of the sentence.

Authors’ Response: Thank you for catching this error. We have corrected it.

Reviewer Comment: Line 369: “This application discloses a two compartment vaporizer wherein the material being vaporized in each compartment could be the same or different, and the vaporization temperature of each compartment could be the same or different” I think this sentence needs to be rephrased into “ ..where the material as well as the temperature can differ at each of the two compartments”.

Authors’ Response: Thank you. The suggested phrasing improves readability and we adopted it. As we attempted to reduce the complexity of the Results section, we ended up removing these phrases completely.

Reviewer Comment: Line 782: I think a “What” is missing between “insight into” and “technology” in order for the sentence to make sense.

Authors’ Response: Thank you. We have made this correction.

Round 2

Reviewer 1 Report

Overall, the authors have significantly improved the quality and scientific soundness of their paper. I am quite satisfied with their responses to my previous queries.

My main suggestion at this point is to double-check all those parts with substantial changes as it seems that there are some sentences that require slight language corrections.

Author Response

Authors’ Response to Review Report 1 (Round 2)

Thank you for reviewing our manuscript following its revision. Your feedback has improved the manuscript and we sincerely appreciate your insights.

Reviewer Comments Related to: Introduction

Reviewer Comment: Overall, the authors have significantly improved the quality and scientific soundness of their paper. I am quite satisfied with their responses to my previous queries.

Authors’ Response: Thank you. Your feedback has strengthened the article.

Reviewer Comment: My main suggestion at this point is to double-check all those parts with substantial changes as it seems that there are some sentences that require slight language corrections.

Authors’ Response: Thank you. We have reviewed the entire manuscript with a read-through, and conducted both a spell-check and grammar check. We have revised the wording of some sentences where the phrasing may have been awkward or ambiguous. We avoided making any structural changes and focused only on editorial changes during the second revision. All changes were marked using “track changes” for easy-cross reference.